# Evaluation of the Caffeine Content in Servings of Popular Coffees in Terms of Its Safe Intake—Can We Drink 3–5 Cups of Coffee per Day, as Experts Advise?

**DOI:** 10.3390/nu16152385

**Published:** 2024-07-23

**Authors:** Regina Ewa Wierzejska, Iwona Gielecińska

**Affiliations:** 1Department of Nutrition and Nutritional Value of Food, National Institute of Public Health NIH—National Research Institute, Chocimska St. 24, 00-791 Warsaw, Poland; 2Department of Food Safety, National Institute of Public Health NIH—National Research Institute, Chocimska St. 24, 00-791 Warsaw, Poland; igielecinska@pzh.gov.pl

**Keywords:** coffee, caffeine, coffee shops, safe consumption, nutritional recommendations

## Abstract

The spreading knowledge of the health benefits of coffee and the development of gastronomy with a wide range of coffees prompt an evaluation of their caffeine content in terms of safe intake. The study analyzed the caffeine content of popular coffees in comparison with recommendations for a safe single dose (200 mg) and daily caffeine intake (400 mg), and guidelines for drinking 3–5 cups of coffee per day. A total of 299 coffee samples from franchise shops and homemade coffees were tested. The “takeaway” coffees had a three times higher mean caffeine content (*p* < 0.005) compared to homemade coffees. Americano coffee was the “strongest” (143 mg caffeine/serving on average), while coffee prepared by pouring hot water over one teaspoon of ground coffee was the “lightest” (23 mg caffeine/serving on average) (*p* < 0.05). Over 200 mg of caffeine per serving was found in 4% of samples. Over 400 mg of caffeine would be consumed by people drinking “on the go” 4–5 servings of many types of coffee, except espresso. In this respect, homemade coffees are safer. Therefore, recommendations on drinking coffee should be more practical, and indicate not only the number of cups, but also the “strength” of various types of coffee, in order to avoid the regular intake of high amounts of caffeine.

## 1. Introduction

Data on coffee’s health benefits have started to grow exponentially [1,2,3,4] since a study showed that regular coffee drinkers are less likely to suffer from type 2 diabetes [5]. As a result, coffee is no longer seen as a risky product, and, on the contrary, experts believe that drinking 3–5 cups of coffee a day fits into the pattern of a health- promoting diet [6,7,8,9,10,11,12]. However, due to the huge differences in portions served (25–470 mL), an even more specific definition of cup size is necessary [13,14,15]. Indeed, the consumer does not know how to understand a cup of coffee. Should a mug of coffee be treated as one cup or as two? Also, perhaps instead of the number of cups, researchers should talk about the volume of coffee consumption as defined by the amount of liquid.

Coffee is a complex mixture of many biologically active substances, but the most characteristic ingredient is caffeine. Although this alkaloid is responsible for the improvement in mood after coffee [16,17,18], its health properties are actually due to other components, which is supported by the positive effects also found in decaffeinated coffee drinkers [7,19,20,21,22]. The composition of coffee is influenced by many factors, dependent and independent of the person making the brew, including the botanical species of coffee, the growing conditions, brewing method, and strength of the brew [14,23,24]. With these multiple determinants in mind, some have reported that the caffeine content of a cup of coffee varies considerably globally; it is 140 mg in Northern Europe, 85 mg in the USA, and only 50 mg in Southern Europe [25]. In the literature, it is, therefore, emphasized that a cup of coffee is neither a well-characterized measure of coffee consumption nor of its components [15,26,27,28]. Furthermore, with the huge growth of chain coffee shops [29], the need to supplement the limited data available on the caffeine content of coffees consumed in town is indicated [14,17,30]. On the other hand, The European Food Safety Authority (EFSA), expresses the view that a meaningful way of assessing caffeine consumption in individual countries requires taking the caffeine content of coffees analyzed on the local market [31]. Some even propose that, in the same way as alcohol units, a conversion factor should be established, whereby different types of coffees, i.e., with varying caffeine contents and serving sizes, should be expressed in coffee units [28].

The effects of caffeine on the body depend on the amount consumed [32]. According to the EFSA position, the safe intake for adults habitually consuming caffeine is up to 400 mg per day, for pregnant women up to 200 mg, and a single dose in adults should not exceed 200 mg (approximately 3 mg/kg body weight in a person weighing 70 kg) [31]. An upper limit of 400 mg of caffeine per day for adults is also set in the USA and Canada [6]. At the same time, it is worth noting that, in light of the evaluation of health claims on food, it is considered scientifically proven that a single intake of at least 75 mg of caffeine increases alertness, which has not been shown for lower doses [33]. Therefore, from the point of view of mental performance, the choice of very light brews is not optimal, and considering all the health aspects of caffeine, coffee containing between 75 and 200 mg of this compound would be the most beneficial.

In regard to the need for a closer characterization of coffee servings from the nutritional side, this study aimed to accomplish the following:−Analyze the caffeine content of popular home-brewed coffees in Poland, as well as coffees from franchised cafés and other places where take-away coffees are purchased;−Identify those types of coffee that contain at least 75 mg but no more than 200 mg of caffeine per serving;−Assess the caffeine content of coffees in terms of a safe, daily caffeine intake (400 mg), in the context of recommendations to drink 3–5 cups of coffee per day.

## 2. Materials and Methods

### 2.1. Study Materials

Due to the types of coffees analyzed, the study consisted of two parts.

#### 2.1.1. Study 1—Commercially Brewed Coffee

The study covered 4 types of the most popular coffees in Poland [29], i.e., espresso, cappuccino, caffè latte/latte macchiato, and Americano, which were purchased in selected franchise shops. These were publicly available, ready-to-drink coffee infusions prepared immediately prior to purchase.

The study was conducted on a group of 208 coffee samples purchased at the following locations:(a)Four popular chain coffee shops (Costa Coffee, Starbucks, Tchibo, and Green Caffè Nero) in the area of Warsaw—sixty-four samples.(b)Three popular chain bakeries (Galeria Wypieków “Lubaszka”, Przystanek Piekarnia, and Putka) in the area of Warsaw—forty-eight samples.(c)Three chain petrol stations (BP—Wild Bean Cafe; Orlen; Shell) in the area of Warsaw—forty samples.(d)Two chain restaurants (McCaffé—McDonald’s and the Metropol hotel restaurant) in the area of Warsaw—thirty-two samples.(e)One chain of small grocery shops (Żabka Polska) in the whole country—twenty-four samples.

All franchise stores, with the exception of two petrol stations, sampled the 4 types of coffee mentioned above. In the case of Orlen and Shell petrol stations, only espresso and Americano coffees were sampled due to the narrower range of products offered. Coffee samples were taken as follows:−Four times at the same outlets in Warsaw, in the months of July, September, October, and November 2021 (applicable to chain coffee shops, BP petrol stations, and McCaffé restaurants), and in the months of June and October 2023 and January and March 2024 (applicable to bakeries and hotel restaurants);−Once in 6 randomly selected locations throughout the country in the period from July–October 2023 (concerning the Żabka franchise shops and the Orlen and Shell petrol stations). Coffee samples were taken by official food control staff as part of monitoring studies in 2023 for the determination of contaminants. The collected samples were also used for the determination of caffeine.

In the case of coffees from surveillance studies (collected in accordance with Regulation (EC) No 333/2007) [34], a coffee sample consisted of 3–6 ready-to-drink coffees in question, depending on the size of the serving. Purchased coffee samples were continuously delivered to the laboratory of the Department of Food Safety of the National Institute of Public Health NIH—National Research Institute. After cooling (to room temperature), the volume of each delivered coffee sample was first measured, and then average laboratory samples were prepared (thoroughly mixed) and their caffeine content determined.

#### 2.1.2. Study 2—Coffee Brewed in the Standardized Laboratory Conditions

The study included instant and ground coffee infusions prepared in the laboratory of the Department of Food Safety using a simple homemade method (hereafter referred to as homemade coffees). The infusions were prepared using coffee samples collected by official food control officers from randomly selected shops and markets/supermarkets throughout the country also as a part of monitoring studies in 2023. A total of 26 samples of instant coffee and 65 samples of roasted coffee (ground or beans) were collected.

These samples were taken in accordance with Regulation (EC) No. 333/2007 [34]. A sample consisted of at least two commercial packages of a product with the same name, from the same producer, from the same production batch, in a quantity of at least 1 kg. The samples taken were delivered to the laboratory of the Department of Food Safety, where they were stored under the conditions indicated on the label until brewing. Two cups of coffee were prepared from each average laboratory sample. A cup of instant coffee was prepared in a very popular way in Poland [29] i.e., by pouring one teaspoon of coffee (2.0 g) over 160 mL of hot but not boiling water and then stirring until the coffee was completely dissolved. In contrast, a cup of ground coffee infusion was prepared by pouring one teaspoon (2.5 g) of ground or pre-ground coffee (beans were ground in a laboratory grinder [final particle size < 300 μm] immediately before brewing) into hot but not boiling water (160 mL), stirring, and then steaming for 5 min. After this time, the infusion was strained from the coffee grounds (i.e., the coffee infusion was decanted). After all the prepared infusions had cooled to room temperature, an average laboratory sample was prepared (two infusions from the same coffee sample were thoroughly mixed), and its caffeine content was determined.

### 2.2. Methods

#### 2.2.1. Determination of Caffeine Content

Caffeine content in coffee infusions was determined by high-performance liquid chromatography using a photodiode array detector (HPLC-DAD) according to an in-house test procedure [35]. The aforementioned method was previously validated and accredited by the Polish Centre for Accreditation to the requirements of EN ISO 17025 (2017). In brief: the coffee infusion was measured into a volumetric flask, diluted five or ten times (depending on the expected caffeine content), and Carrez I and Carrez II solutions were added. The sample was then centrifuged (10,000 rpm; 10 min; approximately 100 °C; MPW-350R; MPW Med. Instruments, Warsaw, Poland), and the clarified supernatant was filtered (PVDF syringe filter; 0.45 μm) and transferred to a chromatography vial.

Caffeine determination was performed using an Alliance 2695 liquid chromatograph with a photodiode array detector (Waters Corporation, Milford, MA, USA) on a Lichrospher RP-18 column (125 mm × 4 mm; 5 μm; Agilent Technologies, Santa Clara, CA, USA). Chromatographic analysis conditions were as follows: mobile phase (water–methanol; 70:30; *v*/*v*), isocratic flow 0.9 mL/min, dispensed sample volume 20 μL, sample temperature 20 °C, column temperature 40 °C, UV detection (λ = 273 nm), analysis time 8 min [35].

Identification of the test compound was based on its retention time and comparison with the spectrum of the caffeine standard. The result is the mean of the three parallel determinations, corrected for recovery.

#### 2.2.2. Calculation of Caffeine Content in a Coffee Serving

To calculate the caffeine content of a serving of coffee, the individual serving size for each type of coffee was used. For ready-to-drink coffee infusions, depending on the place of purchase, the mean serving (min–max) values were, respectively, 34.1 mL (16–60 mL) for espresso, 188.4 mL (102–336 mL) for cappuccino, 206.4 mL (142–312 mL) for caffè latte/latte macchiato, and 225.4 mL (152–420 mL) for Americano. For coffee prepared in the laboratory from instant or ground coffee, a serving (cup) was 160 mL of brew.

#### 2.2.3. Statistical Analysis

The analytical results obtained were evaluated using Dixon’s and Grubbs’ tests in order to eliminate outliers (affected by outliers) and then subjected to statistical analysis using the following statistical packages: Statistica version 6.0 (Statsoft, Inc., Tulsa, OK, USA) and Statgraphics Centurion 18.1.12.

Data on the caffeine content of coffee were presented in mg/serving of coffee, as a mean and as a minimum and maximum value. The significance of differences between the study variables was assessed by one-way analysis of variance (ANOVA), and comparisons were made using the following statistical methods for independent variables:−Student’s *t*-test—when comparing two groups.−Tukey’s test (for equal or unequal sample sizes)—for multiple comparisons.

In all statistical tests, *p* ≤ 0.05 was used as the deciding level for the significance of differences and relationships between the study variables.

## 3. Results

### 3.1. Caffeine Content per Coffee Serving

The caffeine content of a serving of the coffees studied averaged 82.5 mg, varying over a wide range from 12.8 mg to 309.4 mg (Figure 1). A noticeably higher (more than 3-fold), statistically significant (*p* < 0.005) mean content of this compound was found in coffees purchased from franchise shops, compared to homemade coffees (brewed in laboratory).

With regard to the type of coffee, of all the coffees studied, Americano coffee had the highest mean caffeine content, and brews made by pouring hot water over one teaspoon of ground coffee had the lowest (Figure 2). The differences found were statistically significant (*p* < 0.05).

As can be seen from Table 1, the Americano coffee from the bakery and restaurant had a statistically significant (*p* < 0.05) higher amount of caffeine, compared to the other coffees purchased there. Americano coffee from franchise coffee shops and petrol stations contained a significantly higher (*p* < 0.05) amount of this compound compared to espresso, and in the case of the *Żabka* franchise shop, Americano coffee showed a significantly higher amount of caffeine (*p* < 0.005) as compared to cappuccino.

In the case of homemade coffees, a statistically significant amount of caffeine was found to be almost three times higher (*p* < 0.05) in instant coffee infusions than in ground coffee infusions (Figure 2). The caffeine content also differed, depending on the botanical species of coffee bean. Infusions prepared from ground Arabica coffee contained significantly (*p* < 0.0001) less caffeine than infusions made from a mixture of Arabica and Robusta (mean 18.1 mg vs. 25.8 mg/serving).

Place of purchase influenced caffeine content only for espresso coffee. The highest statistically significant (*p* < 0.05) mean content of this compound was found in coffee purchased from the *Żabka* shop, and the lowest was in coffee from franchise coffee shops. It should also be noted that there were differences in the caffeine content per serving of coffee within individual coffee shops (Table 1).

For each type of coffee, there were large discrepancies in caffeine content within the group. The difference between the smallest and largest amounts of the tested component ranged from 3-fold for homemade coffees and more than 4-fold for espresso, caffè latte/latte macchiato, and Americano to almost 7-fold for cappuccino (Table 1).

### 3.2. Caffeine Content in a Coffee Serving between 75–200 mg

Overall, in all the coffee types tested, a caffeine content of 75–200 mg per serving was found in 42.4% of coffee samples (Figure 3). The highest proportion of samples was found in Americano coffee (74.1%), while not a single serving of coffee brewed by pouring water over ground coffee contained this amount of caffeine. In the case of the latter method, all samples contained less than 75 mg of this ingredient per serving of coffee (Table 2).

Very rarely did the caffeine content per serving exceed 200 mg (Figure 3). For the espresso, caffè latte/latte macchiato, home-brewed ground coffee, and instant coffee, no sample contained this amount of caffeine, and the highest proportion of samples (19%) exceeding 200 mg was found in Americano coffee (Table 2).

### 3.3. Caffeine Content in 3–5 Servings of Coffee vs. A Safe Caffeine Intake (400 mg/Day)

On the theoretical assumption that the consumer drinks one type of coffee per day and that there are no other sources of caffeine in the diet, and also considering the mean caffeine content per serving of coffee, a slight excess of the safe intake (by about 8%) would apply to those who drink three servings of Americano. In the case of drinking five servings of coffee such a risk would also exist, in addition to Americano, for those drinking cappuccino (an excess of almost 29%) and caffè latte/latte macchiato (an excess of almost 22%) (Table 3). The magnitude of caffeine intake according to the amount of coffee consumed is shown in Figure 4.

Given the average serving size of each type of coffee and the mean amount of caffeine it contains, the following quantities of coffee would provide 400 mg of this compound: about 180 mL of espresso (5.3 servings), 630 mL of Americano (2.8 servings), 730 mL of cappuccino (3.8 servings), 850 mL of caffè latte/latte macchiato (4.1 servings), 1000 mL of coffee made by pouring water over instant coffee (6.3 cups), and up to 2800 mL of coffee made by pouring water over ground coffee (17.5 cups).

However, considering the maximum caffeine content of the coffee samples, exceeding the safe amount of caffeine could occur much more frequently. Drinking three cups of coffee would result in an excess of caffeine with Americano, cappuccino, caffè latte/latte macchiato and espresso, and drinking five cups would result in an excess of caffeine with almost all types of coffee (Table 3).

## 4. Discussion

The caffeine content of a serving of coffee reported in the literature varies widely and, depending on the value used, the caffeine intake estimated on this basis may be either overestimated or underestimated [30,31]. Some consider a standard cup of coffee to be approximately 240 mL and to contain 100–135 mg of caffeine [36,37,38,39], while others consider an average cup to be 150 mL and to contain 50–90 mg of caffeine [13,31,40,41,42]. In the case of coffees from franchise shops, it is worth noting that most of them are not served in cups, but in paper or glass vessels, usually with a larger capacity due to the portion size.

Regarding the different types of coffee, a study similar to ours was conducted by Jeon et al., who purchased Americanos from coffee shops, fast food restaurants, and bakeries [43]. Coffees from coffee shops were found to have the highest caffeine content (mean 166 mg caffeine/serving), followed by those from fast food restaurants (108 mg/serving) and bakeries (94 mg/serving). In our study, coffees from these outlets should be ranked in reverse order. Americano coffees from bakeries had the highest caffeine content (mean 187 mg/serving), followed by coffees from fast food restaurants (158 mg/serving), and the lowest amount of this compound was found in coffees from coffee shops (122 mg/serving). The average caffeine content of an Americano from all outlets (143 mg/serving) is in line with data published by van Dam et al., who found that a serving of such coffee from coffee shops contains 150 mg of caffeine [10]. Smaller amounts of this ingredient in a serving of Americano are reported by Mitchell et al. (63–126 mg of caffeine) [44]. It is worth noting that Americano coffee served in coffee shops is made with espresso, often double espresso. In our study, the use of double espresso was reported by baristas in eight places where we bought coffee, but in some cases, it was not possible to obtain such data.

In the case of espresso, the mean caffeine content was significantly lower than in the Americano (mean 75 mg/serving), mainly due to the small volume of this coffee (average 34 mL in our study). With regard to espresso coffees from cafés, the result we obtained (mean 57 mg/serving) is in line with that found by McCusker et al. [13] in a serving of espresso from American cafés (58 mg of caffeine in a 30 mL serving). A higher level of this ingredient was found in Scottish espresso from 20 different outlet coffee shops [45], but the results of this study are reported as a median (140 mg/serving, median serving 43 mL), which makes it difficult to compare the data.

Regarding cappuccino coffee, the mean caffeine content in our study (103 mg/serving) is lower than in cappuccinos from Scottish cafés, but again the results of the study are expressed as a median (180 mg/serving) [26]. On the other hand, the result we obtained is in line with the data reported by Mattioli and Farinetti, who found 110 mg/serving [46]. In the case of caffè latte, according to the USDA Food Composition Databases, this coffee contains as much caffeine as a cappuccino (average 86 mg/serving) [47], which is very close to the results of our study, i.e., 97 mg and 103 mg/serving, respectively. Also, in Norway, caffè latte and cappuccino are considered to contain the same amount of caffeine, but at a much lower level—21 mg/100 mL [48], which would mean 50 mg in a 240 mL serving.

The results of our study may be particularly useful with regard to coffee prepared at home by pouring hot water over ground or instant coffee, as this is the most common method used by consumers in Poland (90% of coffee drinkers) [29]. The resulting caffeine content of coffee brewed from ground coffee (mean of 23 mg, in a 160 mL serving) was the lowest of all coffee types tested. This means that such coffees are much “lighter” compared to coffees drunk outside the home. Also, according to the literature, ground coffee infusions made from the same amount of coffee as in our study (one teaspoon = 2.5 g) have a low amount of caffeine, with an average of 36 mg in a 160 mL serving [49] or 46 mg in a 150 mL serving [50]. A much higher amount of this compound (from 199 mg to 241 mg/100 mL) was found in the study by Wołosiak et al. in coffees prepared by pouring hot water over 10 g of ground coffee [51]. In practice, however, this method of brewing is rarely used, as it means using up to four teaspoons of coffee per 100 mL serving. In the case of instant coffee, the caffeine content obtained is typical of the literature data, and it appears that instant coffee is the best characterized in terms of caffeine content compared to all other types of coffee. In our study, a 160 mL serving of coffee brewed from one teaspoon of concentrate contained a mean of 64 mg of this compound, an amount very much in line with the results of Jarosz et al., who obtained a mean of 61 mg from coffee brewed in the same way [49]. Sengpiel et al. report that 100 mL of instant coffee contains 40 mg of caffeine [48] and Fitt et al. found 45 mg [52], which means that a 160 mL cup would contain 64–72 mg of caffeine. According to the International Food Council, a cup of instant coffee (approximately 225 mL) contains 60–85 mg of caffeine [30], and according to Harland, a 140 mL serving contains 66 mg [53]. The higher caffeine content of instant coffee compared to ground coffee may come as a surprise to the average consumer, as Internet blogs suggest that the caffeine content of instant coffee is up to three times lower [54].

Our study confirms the large discrepancies between the minimum and maximum caffeine content of coffee samples, which are repeatedly highlighted in the literature [13,26,45]. The largest difference (almost 7-fold) was found for cappuccino and the smallest (3-fold) was found for homemade coffees. As mentioned above, many factors influence the composition of coffee, one of which is the type of coffee. Robusta coffee contains more caffeine than Arabica [25]. We were only able to obtain data on coffee species for beans or ground coffee (as a commercial raw material), however, as information on coffee species is not mandatory on packaging, although we had such data for 63% of the ground coffee samples tested. Our results are consistent with existing knowledge, as coffee infusions made from Arabica species contained less caffeine than those made from a blend of Arabica and Robusta.

Considering the amount of caffeine per serving of coffee in relation to the dose that is beneficial for alertness in most people (75 mg) [33], more than half of the coffees tested would have to be considered too “light” to be expected to have this effect. These mainly included domestic coffees, for which 89–100% of the samples tested contained less than 75 mg of caffeine per serving, and espresso coffee (59% of the samples). If two teaspoons of coffee are used per serving of homemade coffee, studies suggest that such brews automatically have double the caffeine content (mean 74 mg/160 mL) [49], and, therefore, brewing coffees from a larger quantity of raw material brings them closer to a dose that has a positive effect on the body’s perception. As for a single dose of more than 200 mg of caffeine, which can cause irritability, sleeplessness, or stomach problems and which the EFSA says should not be exceeded [31,55], the risk is low, as less than 4% of all coffee samples were affected. The greatest risk may be for drinkers of Americano, where one in five servings purchased contain more than 200 mg of caffeine.

In the WHO International Classification of Diseases (ICD), symptoms caused by high caffeine consumption are included in the section on mental and behavioral disorders caused by stimulants [56]. In general, publications in this area consider people who consume up to 200 mg of caffeine per day to be low caffeine users and those who consume more than 400 mg to be high caffeine users [57,58]. In our analysis, we assumed that if a consumer ingests only one type of coffee and no other caffeinated products, then given the mean caffeine content of a serving of coffee, the amount of 400 mg could be exceeded by people drinking 4–5 servings of most types of coffee on the go, with the exception of espresso. This would also be supported by literature data on the caffeine content of such coffees in other countries [10,26,43,47]. Home-brewed coffees are safer in this respect, but as mentioned above, these are “light” coffees that may not give the body the feeling of increased alertness that is useful in many life situations. A recent study of the Polish population suggests that Poles are most likely to drink coffee at home (96% of coffee drinkers), opting for traditional brewing methods, but increasingly, Poles, especially young people, are coffee shop customers [29]. Regarding the safety of caffeine consumption in pregnant women, the recommended amount (up to 200 mg of caffeine per day) would be exceeded with two servings of Americano or cappuccino.

In scientific publications on the health benefits of drinking 3–5 cups of coffee, many experts also include the important caveat that caffeine intake should be within a range of up to 400 mg per day [9,11]. When assessing the risk of excessive intake of this compound, it is important to bear in mind that caffeine is not only consumed with coffee in the daily diet, and although coffee is the main source of caffeine in many countries [6,44,59,60], in Poland and the UK, for example, tea provides the most caffeine [61,62,63]. EFSA’s assessment of caffeine intake from all sources showed that of the 13 EU countries, intakes at the 95th percentile higher than 400 mg were found in 7 countries and affected 6–33% of adults [31]. Coffee provided the highest amount of caffeine everywhere, but the authors of the report emphasize that assuming a constant amount of caffeine per serving of coffee in all countries may not accurately reflect the consumption levels. In their view, efforts should be made to use national data, where available, on the caffeine content of different types of coffee. Thus, the results of our study fit well with the stated needs, as in our country such data have so far been incidental. In Poland, which was not included in the EFSA study cited above, it is estimated that 15–20% of adults consume more than 400 mg of caffeine per day, with coffee providing 39–60% of the total daily caffeine pool [58,61,64]. In the USA, caffeine consumption in the age group ≥ 35 years at the 95th percentile exceeded 400 mg and was 420–467 mg/day [44], and in China more than 5% of caffeine consumers intake more than 400 mg per day [65]. If it is assumed that coffee provides 50% of the acceptable caffeine intake, then, based on the results of our study, the daily amount of coffee could be: 90 mL for espresso, 315 mL for Americano, 365 mL for cappuccino, 425 mL for caffè latte/latte macchiato, 500 mL for coffee made by pouring water over instant coffee, and 1400 mL for coffee made by pouring water over ground coffee.

High caffeine intake is not harmless to health and can cause psychomotor agitation, muscle tremors, insomnia, gastrointestinal upset, or tachycardia [56,66]. Consuming more than 1000 mg of caffeine per day is particularly risky [55]. However, it is important to remember that the effects of caffeine on the body are not the same in everyone, due to the rate at which caffeine is metabolized. Carriers of the AA allele of the CYP1A2 gene metabolize caffeine rapidly and may derive greater health benefits from rational coffee consumption than carriers of the AC or CC alleles, who metabolize the compound more slowly [57,67]. Some experts even mention the need for personalized caffeine recommendations, but this is not possible at present and further evidence is needed [6].

The value of our study is enhanced by the large number (299) of coffee samples tested, taken repeatedly from the same outlets or from across the country. This makes the study more comprehensive and the results more reliable. However, a limitation of the study is the lack of detailed information on the preparation of the coffees bought in the coffee shops (e.g., the amount of coffee beans, the brewing time and, in the case of coffees with milk, the amount of milk used). Nevertheless, the authors of the study were not concerned with analyzing the factors that influence the strength of coffee, but with a practical approach that may be important for coffee drinkers. With regard to the safety of the caffeine intake analyzed in the article, it should be considered that this is purely theoretical, as the study did not estimate actual coffee consumption, but was based on recommendations to drink between three and five cups of coffee a day, treating a purchased serving of coffee as one cup.

## 5. Conclusions

Coffees served in franchise shops have significantly higher caffeine levels than homemade coffees. Exceedances of 200 mg of caffeine per serving of coffee were rare, affecting 4% of the coffee samples tested. Bearing in mind that caffeine in the diet comes not only from coffee but also from other products, such as tea and the now-popular energy drinks, and in view of the development of places where coffee can be consumed outside the home, in light of our study, the risk of exceeding 400 mg of caffeine in people who drink several coffees from coffee shops becomes high. Therefore, it does not seem possible to combine drinking 4–5 cups of each type of coffee with a caffeine intake that is considered safe.

This means that recommendations to drink 3–5 cups of coffee for health purposes, mainly because of the valuable polyphenols, need more practical guidance. It would, therefore, be advisable not only to talk about the number of cups, but also about serving sizes, and to promote knowledge of the “strength” of different types of coffee, in order to avoid the regular intake of high doses of caffeine.

## Figures and Tables

**Figure 1 nutrients-16-02385-f001:**
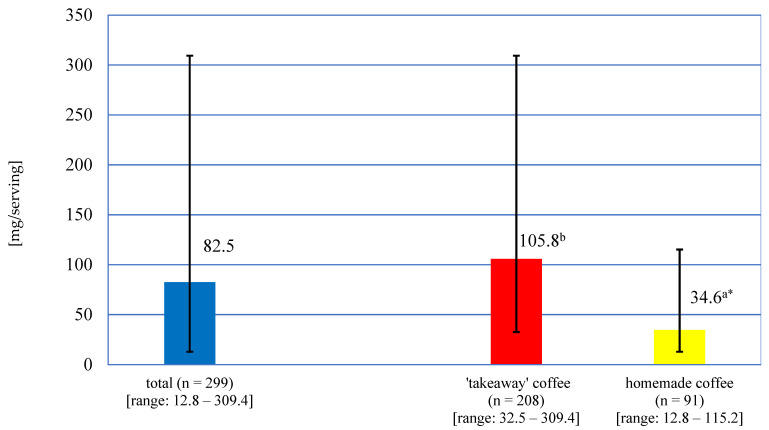
Caffeine content in coffee infusions purchased from franchise shops and homemade (coffees brewed in laboratory); * represents a statistically significant difference (*p* < 0.0001). Values marked with different letters were significantly different.

**Figure 2 nutrients-16-02385-f002:**
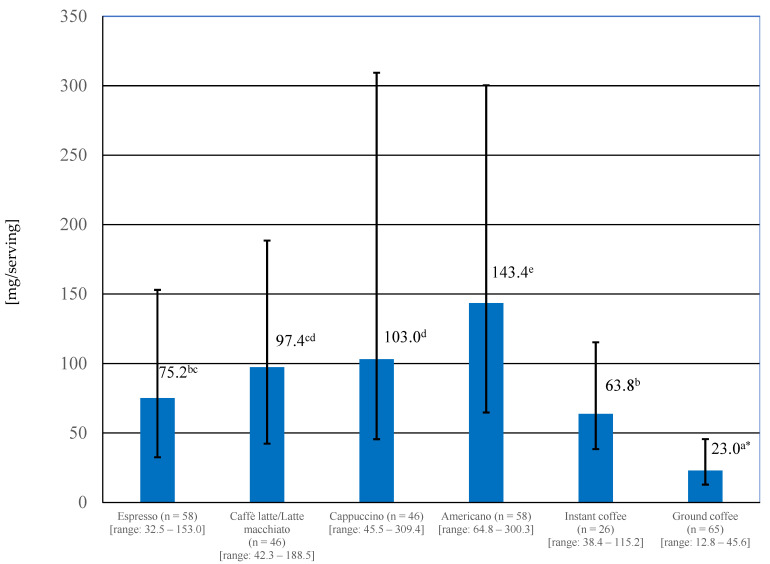
Caffeine content of different coffees purchased in the franchise shop and homemade (coffees brewed in laboratory); * represents a statistically significant difference (*p* < 0.05). Values marked with different letters were significantly different.

**Figure 3 nutrients-16-02385-f003:**
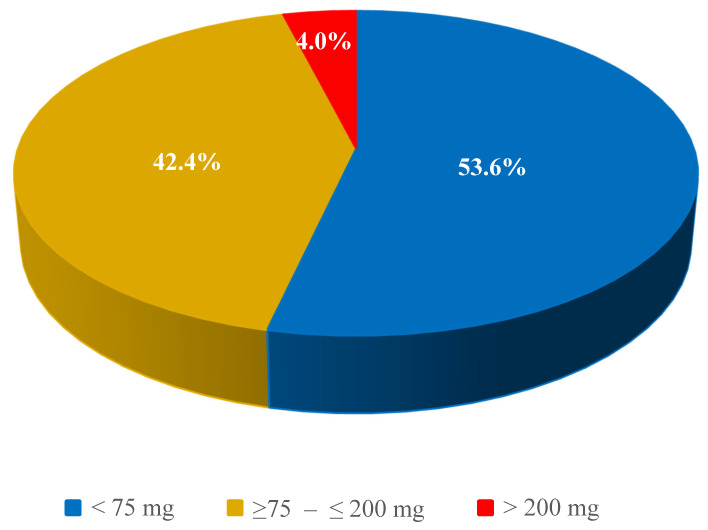
Distribution of the number of coffee samples by caffeine content.

**Figure 4 nutrients-16-02385-f004:**
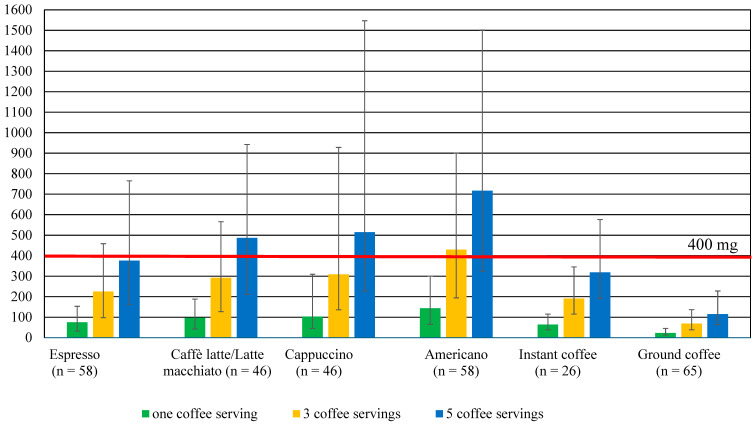
The amount of caffeine provided with coffee at different consumption levels, in relation to the safe caffeine intake in adults (400 mg/day), taking into account the mean, minimum, and maximum caffeine content per serving of coffee.

**Table 1 nutrients-16-02385-t001:** Content of caffeine in different types of coffee collected from franchise shops (in mg/coffee serving) *.

Type of Outlets	Coffee Infusions				
	Espresso (1)	Caffè Latte/Latte Macchiato (2)	Cappuccino (3)	Americano (4)	Difference between Coffee Types *p*-Value
Costa Coffee	59.5 ± 16.1 (32.5–75.4)	70.5 ± 16.4 ^a^*^1^ (46.1–91.5)	69.0 ± 14.2 ^a^*^2^ (52.8–87.4)	72.4 ± 6.1 (64.8–78.4)	
Starbucks	55.2 ± 13.2 (35.7–64.2)	96.1 ± 44.5 ^ab^ (55.4–151.2)	77.0 ± 13.9 ^a^ (56.9–89.0)	171.7 ± 44.8 (119.5–226.4)	
Tchibo	58.3 ± 11.5 (49.1–73.2)	154.2 ± 34.0 ^b^ (107.7–188.5)	211.8 ± 66.6 ^b^ (164.5–309.4)	142.2 ± 86.5 (66.7–219.1)	
Green Caffè Nero	53.4 ± 21.0 (35.5–81.5)	74.6 ± 15.7 ^a^ (52.0–85.9)	95.0 ± 13.7 ^a^ (75.6–106.6)	100.9 ± 27.1 (80.5–140.6)	
Total (all coffee shops)	56.8 ± 14.5 ^A^*^3^ (32.5–81.5)	97.2 ± 43.2 (46.1–188.5)	113.2 ± 67.5 (52.8–309.4)	121.8 ± 60.0 (64.8–226.4)	(1) & (3); (1) & (4) *p* < 0.05
*Galeria Wypieków* “*Lubaszka*”	54.3 ± 2.8 ^a^*^4^ (51.2–56.7)	76.3 ± 20.1 ^a^*^5^ (53.1–88.5)	82.0 ± 12.3 ^a^*^5^ (68.1–91.4)	103.8 ± 8.5 ^a^*^1^ (98.0–113.5)	
*Przystanek Piekarnia*	142.8 ± 11.3 ^b^ (130.6–153.0)	141.0 ± 22.0 ^b^ (115.6–154.1)	153.2 ± 25.4 ^b^ (135.3–182.2)	254.2 ± 42.5 ^b^ (216.5–300.3)	
Putka	74.1 ± 13.2 ^a^ (65.6–89.3)	83.5 ± 17.0 ^a^ (64.3–96.7)	95.2 ± 11.6 ^a^ (82.2–104.5)	204.1 ± 7.8 ^b^ (199.2–213.1)	
Total (all bakeries)	90.4 ± 41.2 ^AB^ (51.2–153.0)	100.3 ± 35.2 (53.1–154.1)	110.1 ± 36.1 (68.1–182.2)	187.4 ± 69.9 (98.0–300.3)	(1) & (4); (2) & (4); (3) & (4) *p* < 0.01
‘*Żabka*’ franchise shop	111.2 ± 15.9 ^B^ (101.0–134.7)	122.9 ± 4.9 (118.0–127.8)	66.4 ± 8.8 (57.1–74.5)	154.9 ± 33.6 (111.0–198.0)	(3) & (4) *p* < 0.005
Fast food restaurant–McCaffé (McDonald’s)	48.0 ± 16.7 ^a^*^5^ (34.1–71.8)	92.0 ± 41.8 (42.6–134.2)	64.5 ± 24.5 (45.5–100.4)	169.3 ± 60.2 (129.4–258.7)	
Hotel restaurant	117.3 ± 30.9 ^b^ (81.9–139.2)	104.5 ± 25.8 (83.1–133.2)	112.5 ± 32.0 (87.4–148.5)	143.1 ± 25.5 (123.1–171.9)	
Total (all restaurants)	77.7 ± 42.8 ^AB^ (34.1–139.2)	97.4 ± 33.8 (42.6–134.2)	85.1 ± 36.0 (45.5–148.5)	158.0 ± 47.2 (123.1–258.7)	(1) & (4); (2) & (4); (3) & (4) *p* < 0.05
BP—Wild Bean Cafe	85.8 ± 15.8 (76.3–109.4)	72.4 ± 25.8 (42.3–105.1)	105.0 ± 35.2 (64.2–150.1)	103.1 ± 32.4 (79.6–149.2)	
Orlen	80.6 ± 10.2 (65.5–90.5)			143.0 ± 53.6 (89.3–226.0)	
Shell	65.4 ± 15.0 (46.2–87.5)			133.3 ± 15.6 (108.6–153.5)	
Total (all petrol stations)	76.2 ± 15.4 ^AB^ (46.2–109.4)	72.4 ± 25.8 (42.3–105.1)	105.0 ± 35.2 (64.2–150.1)	130.2 ± 40.1 (79.6–226.0)	(1) & (4) *p* < 0.0005

* Data are presented as the mean ± SD (min-max) with the Tukey test *p*-value (multiple comparisons) or a Student’s *t*-test *p*-value (comparison of two groups). *^1^ Statistically significant difference (*p* < 0.01) between various types of franchise shops within a group. Values marked with different letters were significantly different. *^2^ Statistically significant difference (*p* < 0.005) between various franchise shops within a group. Values marked with different letters were significantly different. *^3^ Statistically significant difference (*p* < 0.05) between various franchise shops. Values marked with different letters were significantly different. *^4^ Statistically significant difference (*p* < 0.001) between various types of franchise shops within a group. Values marked with different letters were significantly different. *^5^ Statistically significant difference (*p* < 0.05) between various types of franchise shops within a group. Values marked with different letters were significantly different.

**Table 2 nutrients-16-02385-t002:** Distribution of caffeine content in different coffee types.

Type of Coffee	Caffeine Content per Coffee Serving					
	<75 mg		≥75 mg–≤200 mg		>200 mg	
	n	%	n	%	n	%
Espresso (n = 58)	34	58.6	24	41.4	0	0.0
Caffè latte/latte macchiato (n = 46)	16	34.8	30	65.2	0	0.0
Cappuccino (n = 46)	14	30.4	31	67.4	1	2.2
Americano (n = 58)	4	6.9	43	74.1	11	19.0
Ground coffee (n = 65)	65	100	0	0.0	0	0.0
Instant coffee (n = 26)	23	88.5	3	11.5	0	0.0

**Table 3 nutrients-16-02385-t003:** Percentage of caffeine provided with coffee at different consumption levels, in relation to the safe caffeine intake in adults (400 mg/day), taking into account the mean, minimum, and maximum caffeine content per coffee serving.

Type of Coffee	One Coffee Serving	Three Coffee Servings	Five Coffee Servings
Espresso	18.8%	56.4%	94.0%
(n = 58)	(8.1–38.3%)	(24.4–114.8%)	(40.6–191.3%)
Caffè latte/latte macchiato	24.4%	73.1%	121.8%
(n = 46)	(10.6–47.1%)	(31.7–141.4%)	(52.9–235.6%)
Cappuccino	25.8%	77.3%	128.8%
(n = 46)	(11.4–77.4%)	(34.1–232.1%)	(56.9–386.8%)
Americano	35.9%	107.6%	179.3%
(n = 58)	(16.2–75.1%)	(48.6–225.2%)	(81.0–375.4%)
Ground coffee	5.8%	17.3%	28.8%
(n = 65)	(3.2–11.4%)	(9.6–34.2%)	(16.0–57.0%)
Instant coffee	16.0%	47.9%	79.8%
(n = 26)	(9.6–28.8%)	(28.8–86.4%)	(48.0–144.0%)

## Data Availability

Data are contained within the article.

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
