# Peer review of "Evaluation of the Caffeine Content in Servings of Popular Coffees in Terms of Its Safe Intake—Can We Drink 3–5 Cups of Coffee per Day, as Experts Advise?"

_nutrients, 2024, doi:10.3390/nu16152385_

Round 1

Reviewer 1 Report

Comments and Suggestions for Authors

The authors describe an interesting study into the caffeine content of beverages on the Polish market.

The following revisions may be considered:

The major deficit appear the conclusion about so-called "homemade" coffees. These were not actually sampled in households, but prepared in a laboratory setting using a single protocol for preparation. A rationale for this protocol must be provided. 2.5g coffee per 160 ml of water is completely underdosed. Typical guidelines suggest 7g per cup (https://www.coffeecircle.com/en/e/coffee-ratio). The ISO 6668 even suggests 7 g per 100 ml (https://cdn.standards.iteh.ai/samples/44609/99fe9626c50f4f6db934456954f488f2/ISO-6668-2008.pdf).

Hence, this underdosing compared to normal household practices may explain the lower caffeine contents compared to commercial outlets.

Finally, as a remark, caffeine may probably be calculated from bean/powder analyses to cup analysis with reasonable accuracy if the brew ratio is known, because of the excellent water solubility of caffeine.

All in all, I think the results of the comparison between the two study parts need to be toned down.

Some detailed remarks:

Line 27: I would not start a paper with "since the study shows". 

Line 28: Other health benefits include lowered risk for liver cancer (IARC monograph Vol. 116)

Line 95: define Americano coffee (this is diluted espresso?). Is filter coffee not available in Poland?

Line 179 and throughout: suggest to round off the numbers to whole number. The decimals do not make much sense considering the variance.

Discussion: can the results of EFSA be compared in more detail (even if no data for Poland appear to have been used)

Line 68: the only 3-fold difference can be explained by the simplistic protocol of only one brew-ratio. This is basically only the variance of the used coffee beans and does not include the coffee preparation variance.

Author Response

Reviewer 1

The major deficit appear the conclusion about so-called "homemade" coffees. These were not actually sampled in households, but prepared in a laboratory setting using a single protocol for preparation. A rationale for this protocol must be provided. 2.5g coffee per 160 ml of water is completely underdosed. Typical guidelines suggest 7g per cup (https://www.coffeecircle.com/en/e/coffee-ratio). The ISO 6668 even suggests 7 g per 100 ml (https://cdn.standards.iteh.ai/samples/44609/ 99fe9626c50f4f6db934456954f488f2/ISO-6668-2008.pdf).

Hence, this underdosing compared to normal household practices may explain the lower caffeine contents compared to commercial outlets.

According to the information provided in Material and Methods, the coffee was brewed in a laboratory (by the same person), but the brewing method corresponded to how Poles brewed coffee at home. In the light of the publication: Czarniecka-Skubina E. et al. Consumer choices and habits related to coffee consumption by Poles (Int. J. Environ. Res. Public Health 2021, 18, 8, 3948), the use of one teaspoon of coffee (2.0 - 2.5 g) per serving is common in our country, and only less than 4% of people use more than 3 teaspoons. The same amount of coffee was also used by other authors, both in Poland (e.g. Dąbrowska-Molenda M. et al. Analysis of caffeine content in selected types of coffee. Post. Tech. Przetw. Spoż. 2029, 2, 68-71) , as well as in the world (Ludwig I.A. et al. Variations in caffeine and chlorogenic acid contents of coffees: what are we drinking? Food Funct. 2014, 5, 8, 1718-26). Appropriate information has been added in Material and Methods (line 130). In the discussion of the results, we provide information that, according to publications, coffees brewed using two teaspoons of coffee automatically have double the amount of caffeine, so the consumer can easily calculate it. However, in the abstract (line 19) and in results (line 189) was added information that the coffee was brewed with one teaspoon of coffee.

The 7 g of coffee mentioned by the reviewer is a typical amount used to brew espresso coffee [Yildirim S. et al. Use of electrochemical methods to determine the effect of brewing techniques (Espresso, Turkish and Filter coffee) and roasting levels on the antioxidant capacity of coffee beverage. J Food Sci Technol. 2023, 60, 7, 1933-1943;  Grioni S. et al. Espresso coffee consumption and risk of coronary heart disease in a large Italian cohort. Plos One 2015,  10(5):e0126550]. The website mentioned in the review is a internet blog, where there is information about using 7 g of coffee for brewing espresso.

Finally, as a remark, caffeine may probably be calculated from bean/powder analyses to cup analysis with reasonable accuracy if the brew ratio is known, because of the excellent water solubility of caffeine.

Such a calculation of the caffeine content can be made, but we wanted to estimate the caffeine content in coffees offered by cafes, and not in a theoretical way.

All in all, I think the results of the comparison between the two study parts need to be toned down.

As mentioned earlier, information about the use of one teaspoon of coffee was added, which makes the reader aware of what coffees were tested. We hope this explanation is satisfactory for you.

Some detailed remarks:

Line 27: I would not start a paper with "since the study shows". 

The sentence has been corrected [lines: 29-32].

Line 28: Other health benefits include lowered risk for liver cancer (IARC monograph Vol. 116)

Of course, you are right, there is such data in the literature, but we do not write in detail about the risk of particular diseases, but generally that coffee has become a health-promoting product.

Line 95: define Americano coffee (this is diluted espresso?). Is filter coffee not available in Poland?

Americano coffee offered in coffee shops (referred to as American coffee imitation) is espresso-based, often double espresso. Filter coffees are rarely offered in coffee chains. We did not analyze such coffees in this study.

Line 179 and throughout: suggest to round off the numbers to whole number. The decimals do not make much sense considering the variance.

The statistical analysis included values ​​rounded to tenths (the form of obtained analytical results), therefore such values ​​were presented in the manuscript. This is how the authors present the results of caffeine content in coffee infusions in other publications.

Discussion: can the results of EFSA be compared in more detail (even if no data for Poland appear to have been used).

The EFSA study was a very extensive study. In our article, we quoted something that is closely related to the topic of our study, i.e. how many people consume more than 400 mg of caffeine.

Line 68: the only 3-fold difference can be explained by the simplistic protocol of only one brew-ratio. This is basically only the variance of the used coffee beans and does not include the coffee preparation variance.

According to the authors, the difference is not only due to the number of coffee beans but also to the brewing method, mainly pressure because cappuccino in cafes is prepared from espresso coffee. The authors did not want to analyze the factors influencing the caffeine content in coffees, but simply to see how much caffeine is in the coffees consumers drink. We hope this explanation is sufficient for you.

Reviewer 2 Report

Comments and Suggestions for Authors

This article has a theme the problem of caffeine content in different drinks sold throughout Poland. Indeed, the authors start from the hypothesis that since there is no unitary definition of a cup of coffee, it is important to know how much of the alkaloid is usually drunk in a portion. All in the framework of the recommendation to drink 3-5 portions per day, in order to gain health benefits, especially cardiovascular ones.

They tested many samples of coffee types sold in the country in different places, from chains, to petrol stations and also prepared themselves in the laboratory coffee from instant and ground beans. 

Results are thoroughly presented, and in accordance to many other similar results from literature. Indeed, caffeine has many variations, according to preparation method or , mainly, to the botanical type. In this framework, the authors recon that it is very difficult to respect the upper limit of 400 mg per day, or max of 200 mg per one drink of caffeine and to get benefic effects while avoiding intoxication. The discussion chapter compares results with those from many other studies and emphasises the need to formulate in other terms the 3-5 portions per day recommendation. 

The research is very useful , there are a lot of figures assisting us in understanding everything and contributes to the body of knowledge in this domain. Coffee is so widely drunk, that everything connected with its consumption becomes a public health subject.

References are also up to date.

Please, review a little bit the article and correct some minute errors (eg line 27 - the study?!

From the article I do not understand if you got samples from small coffee shops that sell specialty coffee, the kind of coffee shops very popular in Europe, especially among young people. They are usually not from chains, so accessing them during official control is quite unusual. 

Author Response

Reviewer 2

This article has a theme the problem of caffeine content in different drinks sold throughout Poland. Indeed, the authors start from the hypothesis that since there is no unitary definition of a cup of coffee, it is important to know how much of the alkaloid is usually drunk in a portion. All in the framework of the recommendation to drink 3-5 portions per day, in order to gain health benefits, especially cardiovascular ones.

They tested many samples of coffee types sold in the country in different places, from chains, to petrol stations and also prepared themselves in the laboratory coffee from instant and ground beans. 

Results are thoroughly presented, and in accordance to many other similar results from literature. Indeed, caffeine has many variations, according to preparation method or , mainly, to the botanical type. In this framework, the authors recon that it is very difficult to respect the upper limit of 400 mg per day, or max of 200 mg per one drink of caffeine and to get benefic effects while avoiding intoxication. The discussion chapter compares results with those from many other studies and emphasises the need to formulate in other terms the 3-5 portions per day recommendation. 

The research is very useful , there are a lot of figures assisting us in understanding everything and contributes to the body of knowledge in this domain. Coffee is so widely drunk, that everything connected with its consumption becomes a public health subject.

References are also up to date.

Please, review a little bit the article and correct some minute errors (eg line 27 - the study?)

The correction has been made.

From the article I do not understand if you got samples from small coffee shops that sell specialty coffee, the kind of coffee shops very popular in Europe, especially among young people. They are usually not from chains, so accessing them during official control is quite unusual. 

Thank you for raising this topic. The study did not analyze coffee from small cafes, only coffee from franchise shops. Perhaps we will do such a study in the future.

Reviewer 3 Report

Comments and Suggestions for Authors

MANUSCRIPT: 3089648

TITLE: Evaluation of the Caffeine Content of Popular Coffees in the Terms of Its Safe Consumption. Can We Drink 3-5 Cups of Coffee per Day as Recommended by Experts?

The manuscript 3089648 “Evaluation of the Caffeine Content of Popular Coffees in the Terms of Its Safe Consumption. Can We Drink 3-5 Cups of Coffee per Day as Recommended by Experts?” presents an interesting and valuable study in order to show the caffeine content in various types of coffee drinks and show us for each of them how many cups of each of these drinks are safe to drink without exceeding the safe daily intake of caffeine.

This work is well structured, well planned and the research is competently carried out, the methodology was quite adequate for the research.

The results were not subject to appropriate statistical analysis.

The literature is well cited and most of the papers cited (43.3 %) date back to the last five years.

Conclusions presented are in accordance with the results obtained.

Regarding the manuscript presented, I congratulate the authors for the valuable information on the possible consumption of caffeine through various forms of coffee consumption and I have just one recommendation:

1. In Section 2. Methods. Please carefully review the section 2.2.1. Determination of Caffeine Content and consider add the on each instrument used, namely centrifuge and liquid chromatograph : model, producer and its location (Instrument model, Producer, City, State Abbr., Country). Proceed in the same way for all instruments used.

Author Response

 Reviewer 3

The manuscript 3089648 “Evaluation of the Caffeine Content of Popular Coffees in the Terms of Its Safe Consumption. Can We Drink 3-5 Cups of Coffee per Day as Recommended by Experts?” presents an interesting and valuable study in order to show the caffeine content in various types of coffee drinks and show us for each of them how many cups of each of these drinks are safe to drink without exceeding the safe daily intake of caffeine.

This work is well structured, well planned and the research is competently carried out, the methodology was quite adequate for the research.

The results were not subject to appropriate statistical analysis.

The literature is well cited and most of the papers cited (43.3 %) date back to the last five years.

Conclusions presented are in accordance with the results obtained.

Regarding the manuscript presented, I congratulate the authors for the valuable information on the possible consumption of caffeine through various forms of coffee consumption and I have just one recommendation:

In Section 2. Methods. Please carefully review the section 2.2.1. Determination of Caffeine Content and consider add the on each instrument used, namely centrifuge and liquid chromatograph: model, producer and its location (Instrument model, Producer, City, State Abbr., Country). Proceed in the same way for all instruments used.

Appropriate information has been added in the text of the article (lines: 148, 151-152).